# A Study on the Deployment of Mesoscale Chemical Hazard Area Monitoring Points by Combining Weighting and Fireworks Algorithms

**Yimeng Shi [1], Hongyuan Zhang [1,*], Zheng Chen [2], Yueyue Sun [1], Xuecheng Liu [1] and Jin Gu [1]**

[1] Institute of Nuclear Biological and Chemical Defense, People's Liberation Army, Beijing 102205, China
[2] Beijing Chaotu Junke Digital Technology Co., Ltd., Beijing 102205, China
\* Correspondence: zhybeijing@sina.com

**Abstract:** In order to address the problems of redundancy and waste of resources in the deployment of monitoring points in mesoscale chemical hazard areas, we propose a method for the deployment of monitoring points in mesoscale chemical hazard areas by combining weight and fireworks algorithms. Taking the mesoscale chemical hazard monitoring area as the research background, we take the probabilistic sensing model of telemetry sensor nodes as the research object, make a reasonable grid division of the mesoscale monitoring area, calculate the importance of each grid and perform clustering, utilize the diversity of the fireworks algorithm and the rapidity of the solution to solve the monitoring point deployment model and discuss the relevant factors affecting the deployment scheme. The simulation results show that the proposed algorithm can achieve the optimal coverage monitoring for monitoring areas with different importance and reduce the number of monitoring nodes and redundancy; meanwhile, the relevant factors such as the grid edge length, the number of clusters, and the average importance of monitoring areas have different degrees of influence on the complexity of the algorithm and the deployment scheme.

**Keywords:** chemical hazards; fireworks algorithm; k-means clustering; telemetry; monitoring point placement





## 1. Introduction

With the continuous development of modern science and technology, the variety and quantity of various hazardous chemicals are constantly growing, bringing convenience to people while posing more serious chemical threats to social security. First of all, the traditional chemical threat caused by great power competition and regional conflicts of interest has not dissipated with the signing of the Chemical Weapons Convention [1]. As of the 25th anniversary of the entry into the Chemical Weapons Convention in 2022, the progress of chemical weapons destruction is still seriously lagging behind, and there are numerous security risks, including the serious delay in the destruction of chemical weapons in some countries represented by the United States, the slow progress of the disposal of chemical weapons abandoned by Japan in China, and still-unsigned Chemical Weapons Conventions in some countries [2]. Secondly, the international trend of chemical terrorism is intensifying and, compared with other types of terrorist weapons, chemical terrorism is susceptible to being abused because of its characteristics of easy availability of materials, low cost, ease of synthesis, and the suddenness, group catastrophe, concealment, and human damage when performed [3]. According to the statistics of chemical terrorist attacks in the Global Terrorism Index from 1970 to 2015 [4], chemical poison gas has frequently appeared in terrorist attacks in recent years. In addition, the threat of secondary biochemistry caused by chemical accidents is gradually becoming prominent. The consequences of chemical hazard gas diffusion in accidents such as chemical dangerous leakage, coal mine gas explosion, and earthquake secondary toxic gas leakage in chemical parks are highly severe even though the

chemical hazardous gases contribute to the economic and social value of the industry [5]. The chemical hazards produced by hazardous gases can cause fatal casualties and serious social impacts, so monitoring areas of potential chemical hazards has become significant.

The formation of chemical hazard areas and the deployment of monitoring points are related to the content of atmospheric dispersion [6]. According to the scale and size of atmospheric dispersion research, it is generally divided into three categories: micro-scale research ranges from less than one meter to hundreds of meters; small-scale typically ranges from one to several kilometers; and the mesoscale usually ranges from tens of kilometers [7]. There are many ways to deploy monitoring points, and at this stage, the research on the deployment of chemical hazard monitoring points is more focused on the micro-scale and small-scale, and the larger-scale deployment problems can be considered in combination with GIS [8–12]. The traditional monitoring point deployment of chemical hazard areas is usually characterized by the deployment and monitoring of areas where hazards have occurred, which can only achieve fixed-point measurement, and is affected by detection conditions such as ambient temperature and wind speed, and is mainly used for small-scale chemical hazard monitoring areas. Jung [13], for instance, proposed a deployment method for combustible gas monitoring points considering risk indicators, simulated the diffusion results of combustible gas in the combination of different risk factors, and evaluated the risk probability of leakage scenarios to optimize the deployment of monitoring points. When the scope of chemical hazard area is above the medium scale, considering the risk of toxic and harmful gas leakage, long-distance telemetry of dangerous gases has become one of the important research directions of gas monitoring to ensure the life safety of relevant practitioners. Gas telemetry has been developed for decades and does not require direct contact with the target to track, detect, identify, and estimate concentrations in regions [14,15]. The long-distance telemetry technology developed at this stage is divided into passive telemetry technology, such as Fourier transform infrared spectroscopy [16–19], and active telemetry technology, such as differential absorption lidar (DIAL) technology [20–23], Raman scattering spectrum technology [24–28], etc. Long-distance telemetry has non-contact remote detection capability for continuous wave media, which can meet the requirements of remote monitoring, and its strong flexibility is more in line with the requirements of monitoring chemical hazard areas at the medium scale, which can achieve the goal of real-time online remote monitoring and early warning, to a certain extent. In the current research, the research on the location problems of chemical hazard gas monitoring points primarily focuses on the sensor site selection research of small-scale monitoring areas with leakage source determination and diffusion space determination, while the research on the deployment of chemical hazard monitoring points within tens of kilometers of potential leakage sources and monitoring areas needs to be resolved urgently. Therefore, the site selection and efficient monitoring of mesoscale chemical hazard gas monitoring points have become scientific issues to be solved at the national social security level and have important research significance.

In this paper, we mainly study the model construction method of the deployment of mesoscale chemical hazard monitoring points and select the remote telemetry gas sensor as the monitoring equipment. In view of the differences between the layout method of telemetry sensors in mesoscale scenarios and the traditional micro-scale point-taking style sensor deployment methods, as well as the problem of monitoring resource redundancy that tends to occur in mesoscale chemical hazard monitoring, this paper takes typical mesoscale chemical hazard monitoring areas as the research background, establishes a weight-based chemical hazard area monitoring point coverage model, uses the rapidity of fireworks algorithm convergence and the diversity of fireworks operators to solve the model, and discusses the effectiveness and superiority of models and algorithms under different parameter scenarios.

Our contributions are as follows:

- A novel method for constructing a model for the deployment of mesoscale chemical hazard monitoring points considering weights is proposed to achieve the intensive

monitoring of mesoscale chemical hazards through the correspondence between regional importance and coverage, which solves the problem of monitoring redundancy;

- Combined with the existing optimization methods and the characteristics of chemical hazard areas, the deployment optimization method of telemetry chemical monitoring equipment suitable for mesoscale gas diffusion monitoring and early warning is found, and the influence of each parameter on the model and algorithm is discussed;
- This method can be applied to other mesoscale monitoring models, and the solution algorithm for the deployment of mesoscale chemical hazard monitoring points can also be improved by combining with other heuristic algorithms.

The study is divided into six parts, and the rest of the article is organized as follows. Section 2 describes the work of coverage location problems and fireworks algorithms. The construction method of the mesoscale chemical hazard area monitoring point deployment model considering weights is detailed in Section 3. Section 4 describes the fireworks algorithm used for model solving. In Section 5, experiments are utilized to confirm the feasibility of the model construction approach and solution algorithm, and the impact of various parameters on the findings of the monitoring point layout is discussed. Section 6 is the conclusion.

The workflow diagram is shown in Figure 1 below:

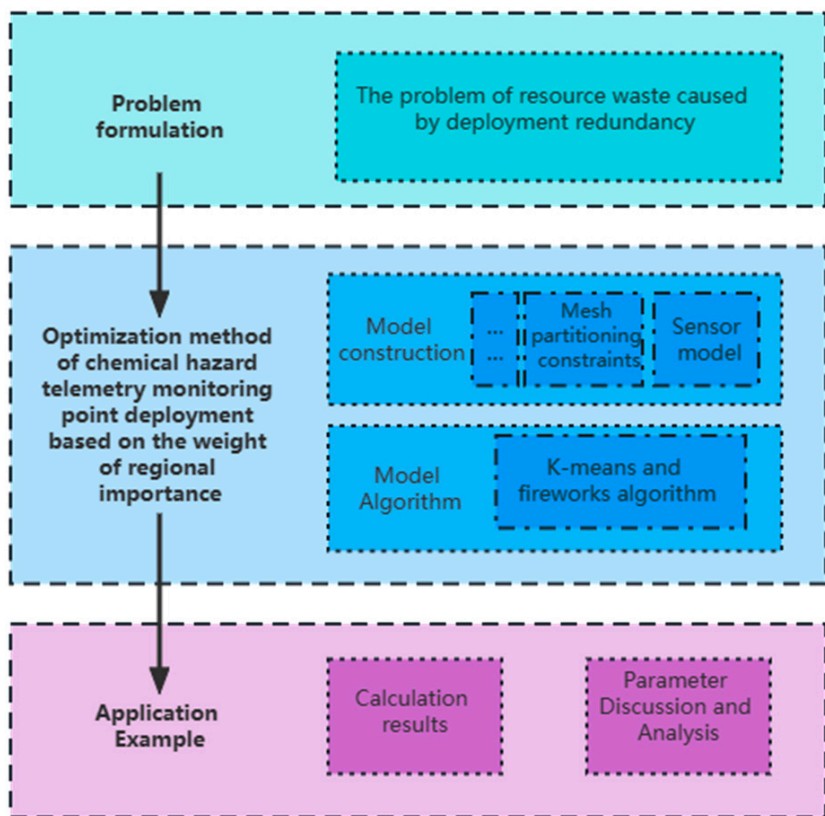

**Figure 1.** Flowchart of the research on the deployment of monitoring points in mesoscale chemical hazard areas by combining weight and fireworks algorithms.

## 2. Related Work

It is inevitable that there will be areas within the monitoring range that lack monitoring significance when the monitoring area is broad, but if the entire monitoring area is laid out indiscriminately, it is easy to squander resources. In order to achieve the purpose of monitoring key areas, it is necessary to further refine the monitoring areas, calculate the importance of the refined areas according to the scientific quantitative model of the importance ranking of chemical hazard monitoring areas, and select the most important areas in the monitoring area to cover the monitoring points and choose the site of the

monitoring points, so as to efficiently and accurately detect the gas leakage information of the chemical hazard area. Coverage location problems [29] include location set covering problem (LSCP) [30–34] and maximum coverage location problem (MCLP) [35,36]. Among them, LSCP is committed to establishing the fewest site selection points to cover all demand points under the premise of covering the specified requirements, and transforming the location problem into a problem of minimizing the number of required site selection points. For example, ref. [37] attempted to apply an advanced location allocation model to locate and allocate disaster victims during floods, using LSCP as an allocation model to determine the minimum number of relief centers required to cover all evacuees or affected populations within a specific distance. Ref. [38] used two optimization models, the LSCP model and the MCLP model, to solve the location problem, and found the best location by comparing the corresponding methods and results of different models to optimize the site selection of the location problem of the ready-mix concrete plant in Thailand. Ref. [39] utilized models such as LSCP to minimize disaster risk by determining the minimum number of emergency evacuation centers required to serve all points of need within a specified coverage distance or itinerary.

It is difficult to accurately calculate the complex coverage location problem, and so it is necessary to solve the location model with the aid of mathematical methods and computer technology, and the optimization algorithm. In recent years, many scholars have studied heuristic algorithms such as genetic algorithm [40], ant colony algorithm [41], firefly algorithm [42] and fireworks algorithm [43] to determine the approximate optimal solution of the problem. The fireworks algorithm is a special swarm intelligence algorithm, inspired by the phenomenon of fireworks explosions that can radiate around, introducing random factors and selecting strategy methods to generate a feasible solution by performing an explosive global and local search of the problem, which solves quickly and achieves remarkable global optimization capabilities [44–48]. In recent years, many scholars have used fireworks algorithms to solve the problem of sensor deployment. For example, Tian [49] designed a multi-sensor network optimization model based on the fireworks algorithm for the optimization problem of sensor network deployment in the modern battlefield, using three aspects of total area coverage, common viewing parameters of key targets and sensor resource utilization as evaluation indicators. The simulation results showed that the fireworks algorithm solves quickly, which can make the total area coverage and common viewing parameters of key areas exceed 90%, and the utilization rate of sensor resources is high, which is conducive to improving the coverage of key areas of the battlefield. Gui [50] proposed a fault diagnosis method based on the fireworks algorithm optimization convolutional neural network algorithm to accurately diagnose faulty sensor nodes. They utilized the self-regulation mechanism of global and local search capabilities of the fireworks algorithm to optimize the weights and biases of convolutional neural networks, so as to solve the problem of limited judgment and convergence speed of convolutional neural networks. It can be seen that it is feasible to solve the model by using the convergence speed of the fireworks algorithm and the diversity of fireworks operators.

## 3. The Deployment Model for Mesoscale Chemical Hazard Area Monitoring Point Considering Weight

When constructing a mesoscale chemical hazard area monitoring a point deployment model considering weights, the sensing model of sensors at monitoring points is initially considered, and the exponential decay probability sensing model is selected as the sensing model of monitoring points; a suitable coverage location model, or the LSCP model, is selected according to the monitoring requirements; then, the mesoscale area to be monitored is gridded, and a suitable grid is selected by combining the specific characteristics of the deployment model to achieve a balance between accuracy and computational, and the importance of each grid after division is calculated; the grids are clustered into regions of different importance, the importance of the regions is also obtained, a relationship between

the importance of the regions and the coverage threshold is defined, and, finally, the model is constructed.

### 3.1. Sensing Model

In the study on sensor deployment problems, the most-used sensing model is the Boolean sensing model, and in addition to this, there are probability sensing models, oriented sensing models, etc. [51]. The exponential decay probability sensing model among the perceptual models is chosen as the sensor model in this paper to be closer to reality. Assuming that the monitoring probability $f(s, p)$ that the point $p(x_p, y_p)$ in the chemical hazard area is detected by the sensor at the monitoring point, $s(x_s, y_s)$ satisfies Equation (1).

$$f(s, p) = \begin{cases} e^{-\lambda d(s,p)}, & d(s, p) \leq R_s \\ 0, & d(s, p) > R_s \end{cases} \tag{1}$$

where $\lambda$ denotes the parameter related to the physical characteristics of the sensor; $R_s$ is the maximum sensing radius of the monitoring point; $d(s, p)$ denotes the Euclidean distance between the monitoring point $s(x_s, y_s)$ and the monitored point $p(x_p, y_p)$, as in Equation (2).

$$d(s, p) = \sqrt{(x_s - x_p)^2 + (y_s - y_p)^2} \tag{2}$$

The probability $f(p)$ that point $p(x_p, y_p)$ is monitored by the set S of monitoring points satisfies Equation (3).

$$f(p) = 1 - \prod_{i=1}^{k} (1 - f(s_i, p)) \tag{3}$$

It is known that the minimum perception probability of a monitoring point is $f_{min}$. The more sensitive the sensor is, the smaller the value of its minimum perception probability $f_{min}$ is, and vice versa. When the monitoring probability $f(p)$ of the monitoring point set S for point $p(x_p, y_p)$ is greater than $f_{min}$, it is defined that the point can be monitored and the monitoring effect $F_c(p)$ is marked as 1. The monitoring effect $F_c(p)$ is expressed as 0/1 as follows in Equation (4).

$$F_c(p) = \begin{cases} 1, & f(p) \geq f_{min} \\ 0, & f(p) < f_{min} \end{cases} \quad f_{min} \in [0, 1] \tag{4}$$

### 3.2. Mesh Generation and Mesh Importance

The discretization of the area to be monitored into a series of grid points is a common method in the deployment of sensing network nodes, and general shapes used for mesh generation are triangles, squares, square hexagons, etc. [52]. This paper adopts the square mesh generation method, and meshed chemical hazard areas with $R_g$ as the edge length. The importance $\omega_i$ of mesh generation is calculated through the importance ranking model of chemical hazard areas.

We set the central point of each grid as the discrete alternative points when conducting the research on deployment problems of monitoring point, and assigned importance weight value within each grid to the central points. If the grid edge length is smaller, the mesh generation is denser, the accuracy is higher and the calculation amount is larger. Additionally, the grid statistical method is used to calculate area coverage rate. When the detection range of the monitoring point completely covers the grid, this grid is defined to be monitored, and so the edge of grid $R_g$ satisfies the constraint relation Equation (5):

$$e^{-\lambda \frac{R_g}{\sqrt{2}}} \geq f_{min} \tag{5}$$

After arranging, we have Equation (6):

$$R_g \leq -\frac{\sqrt{2}\ln f_{min}}{\lambda} \tag{6}$$

where $f_{min}$ is the minimum perceived probability of monitoring points.

Subsequently, the importance of each grid in the monitoring area is calculated based on the quantitative model of the importance ranking of the chemical hazard monitoring area, and the importance value $\omega_i$ of each grid is obtained $i$.

### 3.3. Regional Importance and Coverage Threshold

In the case of limited monitoring nodes, priority should be given to the coverage of important areas within the monitoring region. The requirement of the corresponding coverage for different regional importance is also disparate. For instance, the requirement of coverage rate in sparsely populated areas of no significant value is far below that of geographical locations where command departments are located, in which deploying the monitoring point in non-essential areas not only occupies the transmission of communication channels, but also is a waste of deployment resources. k-means clustering of each grid importance $\omega_i$ is performed to form regions to be monitored with different importance. The mean value of importance $\omega_{ave}$ of the whole region is used to calibrate the importance distribution of this monitoring region.

We used the k-means clustering algorithm to cluster a larger number of meshes to form several clusters, i.e., regions of different importance, and obtained the importance $\omega_{ci}(0 < i < C)$ of each region and the number $A_{\omega_{ci}}$ of meshes in each region. According to the different importance of the regions formed after clustering, the corresponding minimum coverage, i.e., the coverage threshold $P_{\omega_{ci}}$ is introduced, and its calculation formula is as in Equation (7).

$$P_{\omega_{ci}} = \frac{1}{2}sin\left(\pi\omega_{ci} - \frac{\pi}{2}\right) + \frac{1}{2} \tag{7}$$

The higher the importance of the area, i.e., the greater the $\omega_{ci}$, the greater the corresponding coverage, i.e., the larger the required coverage threshold value $P_{\omega_{ci}}$.

### 3.4. The Definition of Chemical Hazard Monitoring Points Deployment Model Considering Weight

In the $m \times n$ chemical hazard monitoring area A, dividing the grid with $R_g$ as the radius, the coordinates of each target point of each grid importance are calculated as $p(x_p, y_p)(0 \leq x_p \leq m, 0 \leq y_p \leq n)$. Assuming that K wireless sensor monitoring nodes are deployed, denoting as S, each node coordinate is $s(x_s, y_s)(0 \leq x_s \leq m, 0 \leq y_s \leq n)$. The largest sensing radius of sensor is $R_s$ and its communication radius is $R_c$, which we have known $R_c > 2R_s$, so there are no communication problems by default.

When there are two or more monitoring points $s_i(x_{si}, y_{si})$ and $s_j(x_{sj}, y_{sj})$ in the monitoring area at the same time with monitoring probabilities $f(s_i, p)$ and $f(s_j, p)$ greater than $f_{min}$ for mesh point $p(x_p, y_p)$, it is defined that the point $p$ is repeatedly monitored. The redundancy $F_r(s_i, p)$ of monitoring point $s_i(x_{si}, y_{si})$ to point $p(x_p, y_p)$ is marked as 1, and, similarly, $F_r(s_j, p)$ is also 1, as in Equation (8), and the redundancy coverage $F_r(p)$ of point $p(x_p, y_p)$ under the monitoring point set S is shown in Equation (9):

$$F_r(s_i, p) = \begin{cases} 1, & f(s_i, p) \geq f_{min}, f(s_j, p) \geq f_{min} \\ 0, & otherwise \end{cases} \tag{8}$$

$$F_r(p) = \sum_{i=1}^{k} F_r(s_i, p) - 1 \tag{9}$$

The coverage $P_{cov.\omega_{ci}}$ of the monitoring area with importance $\omega_{ci}$ is calculated by the following Equation (10).

$$P_{cov.\omega_{ci}} = \frac{S_{area.\omega_{ci}}}{A_{area.\omega_{ci}}} = \frac{\sum_{i=1}^{m \times n} F_c(p_i)}{A_{\omega_{ci}}} \tag{10}$$

where $S_{area.\omega_{ci}}$ is the total area of all monitored target points at that importance; $A_{area.\omega_{ci}}$ is the total area of the monitored area with importance $\omega_{ci}$; $F_c(p_i)$ is the monitoring effect of the set S of monitored points on point $p_i$; $A_{\omega_{ci}}$ is the number of meshes with importance $\omega_{ci}$.

The calculation formula for the coverage redundancy $P_{rec}$ of this safeguard area is as follows:

$$P_{rec} = \frac{S_{rea}}{S_{area}} = \frac{\sum_{i=1}^{m \times n} F_r(p_i)}{\sum_{i=1}^{m \times n} F_c(p_i)} \tag{11}$$

$S_{rea}$ denotes the total area of re-monitored target points; $S_{area}$ is the total area of all monitored target points.

To construct a model for the deployment of chemical hazard monitoring points considering the weight, with the coverage of each important region as the constraint, to reduce the redundancy of regional coverage and the number of deployed monitoring points that is the optimization objective, the formula is Equations (12) and (13).

$$minQ = K + 10 \times P_{rec} \tag{12}$$

$$s.t \; P_{cov.\omega_{ci}} \geq P_{\omega_{ci}}, i \in (0, C] \tag{13}$$

where $K$ is the number of sensor monitoring nodes in the coverage area; $P_{rec}$ is the coverage redundancy of this monitoring area; $P_{cov.\omega_{ci}}$ represents the coverage of the monitoring area with importance $\omega_{ci}$; $P_{\omega_{ci}}$ denotes the coverage threshold of the monitoring area with importance $\omega_{ci}$; $C$ is the k-means clustering number of clusters.

## 4. Model Solution Algorithms

### 4.1. k-Means Clustering Algorithm and Fireworks Algorithm

Based on the grid importance values of the chemical hazard monitoring area in the previous chapter, the k-means clustering algorithm was used to recluster the importance into several clusters to form the importance level of the monitoring point deployment. k-means clustering algorithm is the most familiar clustering algorithm, which assigns each piece of data to the nearest cluster in the cluster represented by the center point, given K values and K initial cluster centers. After all the data is allocated, the center of this type of cluster is calculated according to a class cluster in a data center, and then the procedures of assigning data and updating class cluster centers are iteratively performed until the cluster center point changes little or reaches the specified number of iterations.

There are many types of heuristic algorithms [53]. Fireworks algorithm is an intelligent optimization algorithm that simulates the explosion process of fireworks to solve complex optimization problems [54]. Fireworks algorithm uses the new generation of fireworks and variant fireworks which are produced by each generation of fireworks and its explosion to construct solution space determines the feasibility of solution through adaptation value, and obtains the optimal solution after several iterations, which consist of explosion operator, mutation operator, mapping rule, and selection strategy [49].

### 4.2. Middle-Scale Chemical Hazard Monitoring Point Model Solution Steps Considering Weights

Based on the above introduction of k-means clustering algorithm and fireworks algorithm, this paper designs a solution algorithm for mesoscale chemical hazard monitoring point deployment coverage model considering regional importance, and the specific steps are as follows:

Step 1: (k-means clustering) randomly select K samples as the initial cluster center;

Step 2: for the importance data of each grid, calculate the distance to the center of each type of cluster, find the center of the closest cluster, and assign the data to this cluster;

Step 3: recalculate the position of the center of each type of cluster;

Step 4: Determine whether the center position of the cluster has changed. If it changes, repeat the steps of iterating Step 2–Step 3; if it remains the same, then output the result and obtain the clustered clusters that are different importance areas, and the obtained cluster center is the importance value of each region, and the regional importance value is converted into regional coverage by Formula (7) for the calculation of the fitness value of the fireworks algorithm;

Step 5: (fireworks algorithm initialization) initialize various parameters: set the number of explosions M, the number of fireworks, and the maximum number of iterations; initialize the fireworks position, randomly generate some fireworks in a specific solution space, and each firework $x_i$ represents a solution;

Step 6: Calculate the fitness value $f(x_i)$ of each firework according to the optimization objective function to judge the quality of the fireworks. According to Formula (14)–(16), the fireworks explosion amplitude radius $A_i$ and the number of fireworks of the new generation $B_i$ were calculated. According to Equation (15), the number of daughter operators produced by the explosion of fireworks operators is limited to ensure population diversity and optimization speed.

$$A_i = A \times \frac{f(x_i) - V_{min} + \varepsilon}{\sum_{i=1}^{N}[f(x_i) - V_{min}] + \varepsilon} \tag{14}$$

$$B_i = B \times \frac{V_{max} - f(x_i) + \varepsilon}{\sum_{i=1}^{N}[V_{max} - f(x_i)] + \varepsilon} \tag{15}$$

$$B_i = \begin{cases} round(\alpha B) \ (B_i < \alpha m) \\ round(\beta B) \ (B_i < \beta m) \\ round(B_i) \ otherwise \end{cases} \tag{16}$$

$A$ is a constant parameter that controls the explosion radius of the fireworks operator; $L_i$ represents each firework in the current population; $V_{min} = min f(x_i)$ is the minimum fitness of fireworks operators in the current population; $V_{max} = max f(x_i)$ is the maximum fitness of the fireworks operator in the current population; $\varepsilon$ is the minimal constant value that must be set to avoid zero denominator during algorithm calculation; $N$ is the number of total fireworks operators; $B$ is a constant parameter that controls the number of offspring fireworks produced by the explosion of the fireworks operator; round( ) is the rounding function; $\alpha$. $\beta$ is constant and $\alpha < \beta < 1$;

Step 7: Generate explosive sparks and Gaussian sparks. Operate displacement to fireworks according to Formula (17); to ensure population diversity and avoid falling into the local optimal solution, a Gaussian mutation operator with high burst is added, and some fireworks operators are randomly selected to perform Gaussian mutation according to Formula (18). Calculate the fitness value $f(x_i)$ for all sparks and update the global optimal solution; during the operation, the offspring fireworks operator that exceeds the solution domain space is transferred to the solution space according to the mapping rules of Equation (19).

$$\Delta x_i^k = x_i^k + round(0, A_i) \tag{17}$$

$$\Delta x_i = x_i g \tag{18}$$

$$x^k = x_{min}^k + \left| x^k \right| \% \left( x_{max}^k - x_{min}^k \right) \tag{19}$$

where $g$ is a random number that obeys a Gaussian distribution with a mean of 1 and a variance of 1; $x^k$ is the position information of the fireworks operator beyond the solution space in k-dimension; $x_{max}^k$ and $x_{min}^k$ are the upper and lower boundary information of the solution domain space in the kth dimension, respectively; % represents the operation of the modulus;

Step 8: Apply the selection strategy to Next Generation Fireworks for selection. According to Formula (20), the high-quality operator part is selected from the operator set generated by the explosion and mutation process by the roulette wheel method based on the Euclidean distance as the parent generation of the new generation fireworks algorithm.

$$P(x_i) = \frac{R(x_i)}{\sum_{j \in K} R(x_i)} \tag{20}$$

where $R(x_i)$ is the sum of the distances between the individual $x_i$ of the fireworks operator and all other operators in space; $P(x_i)$ is the probability that the individual fireworks $L_i$ will be retained;

Step 9: Determine whether the termination conditions are met. If it is satisfied, stop the search and print the results. Otherwise, return to Step 6.

Through the solution process of Step 1 to Step 9 above, the optimal solution of the mesoscale chemical hazard monitoring point location optimization model can be obtained, so as to monitor the mesoscale chemical hazard area efficiently and reliably, and issue accurate and timely early warning when chemical hazards occur in important areas, providing reliable technical support for scientific emergency measures.

## 5. Experiment Analysis

The typical deployment scene of chemical hazard area monitoring points has been designed to verify the effectiveness of deployment models and its solving algorithms. The experimental scene is a 50 × 50 km chemical hazard area to be monitored, the perceptual radius of monitoring nodes is 5 km, the related parameter of sensor physical features is $\lambda = 0.5$, the minimum perceptual probability of monitoring point is $f_{min} = 0.2$, and $R_g$ km is used as the edge length to divide the square grid to calculate the importance $\omega (\omega \in [0, 1])$ of each grid through the importance ranking model of chemical hazard areas and regional importance mean value $\omega_{ave}$, and then the k-means clustering is applied for the calculated importance of each grid cell with the number of clusters C.

The variation of edge length, number of clusters, and importance distribution can not only change the importance of each grid in chemical hazard areas, but also influence the deployment location, number of monitoring nodes, etc. In order to ensure the accuracy of experiment data, we chose the mean value of experiment data for 100 times as the result. The execution time of the choosing process, the number of nodes, and redundancy are used as the measurement indexes of the algorithm's pros and cons.

### 5.1. Effect of Grid Edge Length

The effect of different grid edge lengths on the execution time of algorithms, the number of nodes, and redundancy and the variation of a grid's radius directly affects the changes of alternative points and further causes changes in their node deployment, and even in the amount of computation when the monitoring areas remain unchanged. The monitoring areas are set unaltered in the experiment, the grid edge length $R_g$ is 1, 2, 3, 4 respectively, and other parameter settings are: number of clustering C = 3, $\omega_{ave} = 0.3$.

When the number of iterations is 500, the optimal results of each experiment are shown in Figure 2:

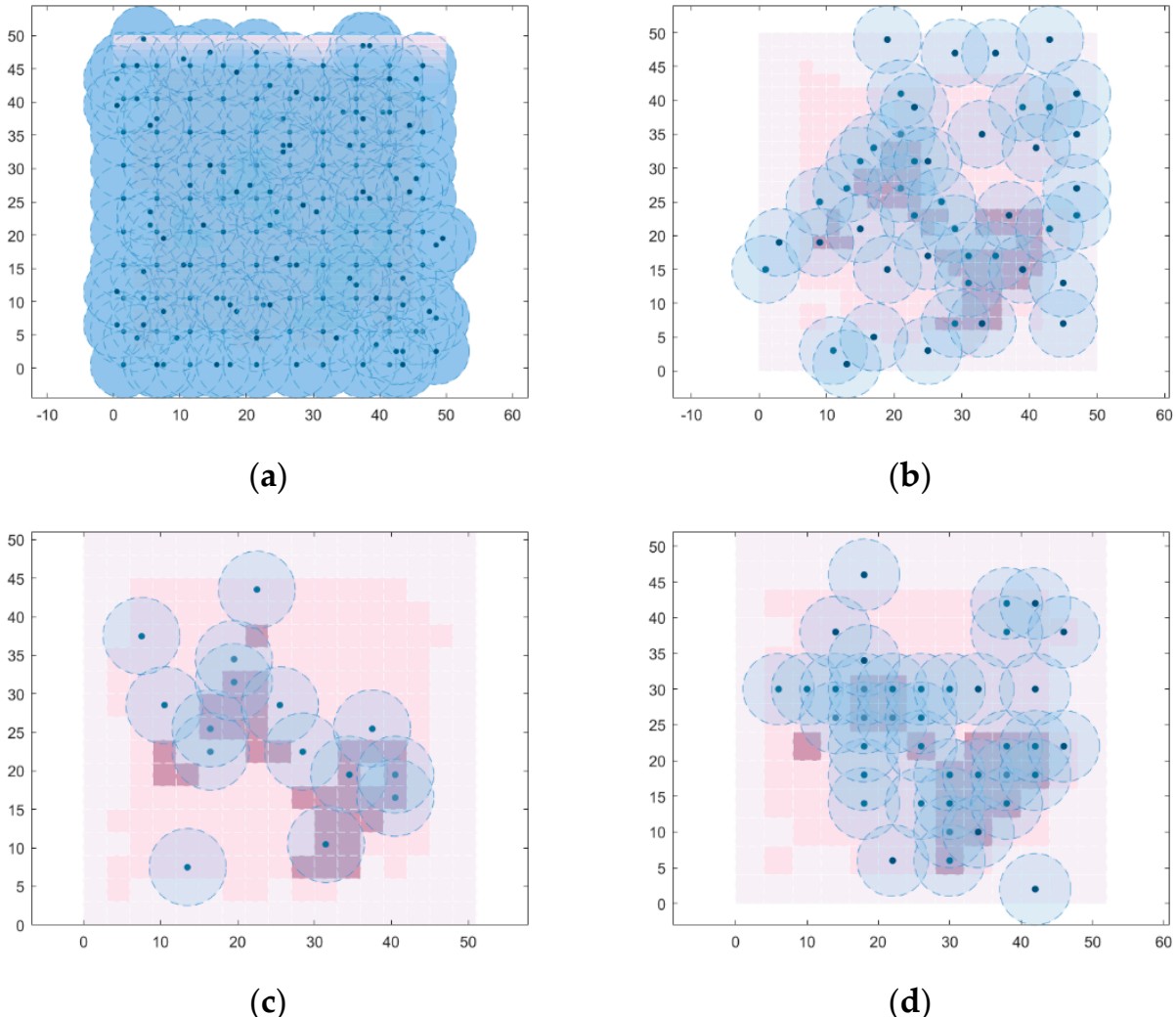

**Figure 2.** Calculated results of monitoring point deployment with different grid edge lengths (unit: km) (In the figure: pink grids indicate different importance areas, blue dots represent monitoring point locations, and blue circles represent monitoring range of monitoring points.): (**a**) grid edge length Rg = 1 km, K = 188, Prec = 1.84, Time = 61,969.814 s. (**b**) Grid edge length Rg = 2 km, K = 45, Prec = 0.039, Time = 4021.676 s. (**c**) Grid edge length Rg = 3 km, K = 15, Prec = 0.02, Time = 996.881 s. (**d**) Grid edge length Rg = 4 km, K = 39, Prec = 0, Time = 430.556 s.

From the analysis of experiment results, we find that grid edge lengths not only affect program run time, but also largely influence the redundancy and number of monitoring points, and the specific results are arranged as shown in Figure 3:

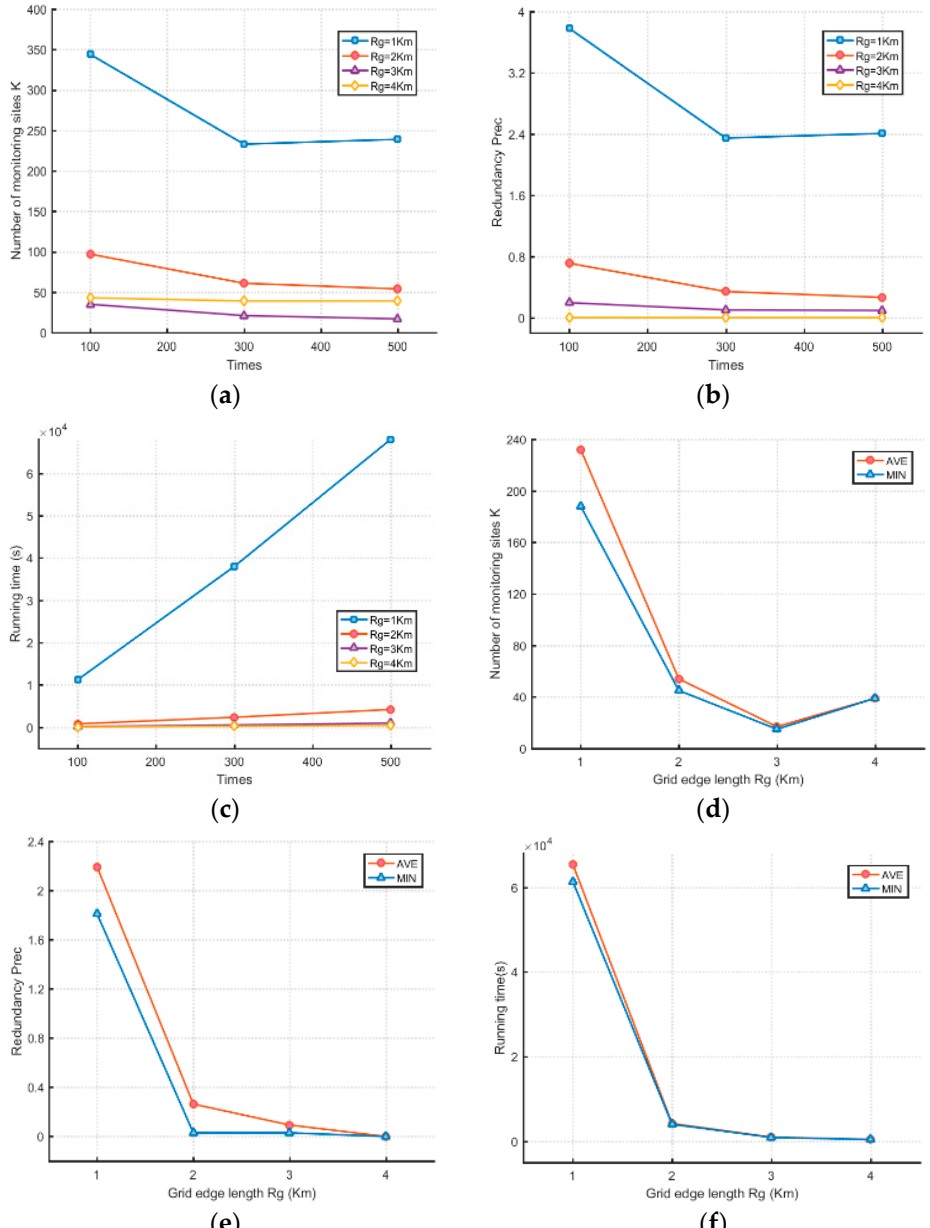

**Figure 3.** Parameters related to the deployment of monitoring points with different grid edge lengths: (**a**) effect of grid edge length on the number of monitoring points. (**b**) Effect of grid edge length on redundancy. (**c**) Effect of grid edge length on running time. (**d**) Mean vs. minimum values for monitoring points with different grid edge lengths. (**e**) Mean vs. minimum of redundancy for different grid edge lengths. (**f**) Mean vs. minimum runtime for different grid edge lengths.

From the above Figure 3, we can find that: (1) as the number of iterations increases, the number of monitoring points and redundancy gradually decrease and stabilize, and the smaller the edge lengths of the grid, the faster the decreasing gradient; (2) as the grid edge lengths increases, the redundancy decreases and the running time rises, and meanwhile, the overall change shows an initial fast followed by a slow trend, and then gradually tends to stabilize; (3) as the increasing of grid edge lengths after the iteration stabilizes, the variation trends of the mean and minimum values of the number of monitoring point, redundancy, and running time are consistent, in which the number of monitoring points displays an initial decrease followed by a rising trend, and redundancy and running time show a fast and then slow downward trend; (4) a comprehensive analysis shows that grid edge lengths have a vital influence on the deployment scheme of monitoring points, in

which the number of monitoring points is minimum when the grid edge length is 3 km, which is 92.6% less than when the edge length is 1 km, the redundancy is 95.7% less, and the running time is 98.5% less, so the comprehensive monitoring effect is optimal.

### 5.2. Effect of the Number of Bunches of Clustering

The effect of the number of bunches of clustering on program running time, the number of nodes, and redundancy under the same monitoring areas. The importance of each grid remains unchanged in the experiment, the number of clustering bunches are 3, 4, 5, respectively, and other parameter settings are: grid edge length $R_g$ = 3 km, $\omega_{ave}$ = 0.3.

When the number of iterations is 500, the optimal results of each experiment are shown in Figure 4:

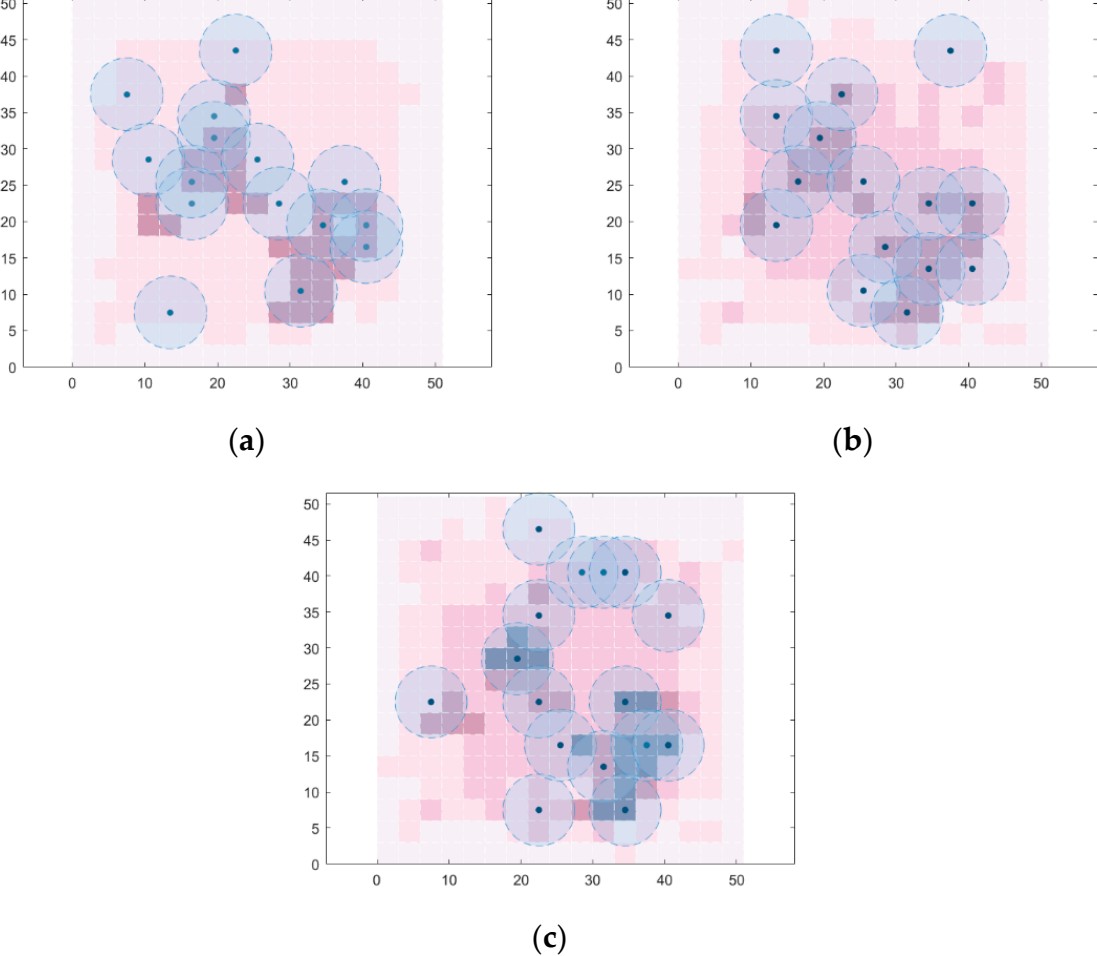

(a)

(b)

(c)

**Figure 4.** The calculation result of monitoring point deployment in different clustering bunches (unit: km) (In the figure: pink grids indicate different importance areas, blue dots represent monitoring point locations, and blue circles represent monitoring range of monitoring points.): (**a**) The number of clusters C = 3, K = 15, Prec = 0.2, Time = 996.881 s. (**b**) The number of clusters C = 4, K = 15, Prec = 0.2, Time = 1036.814 s. (**c**) The number of clusters C = 5, K = 16, Prec = 0.22, Time = 1117.41 s.

To further illustrate the effect of the number of clusters on the experimental results, the specific results of the experiment are organized as shown in Figure 5 below:

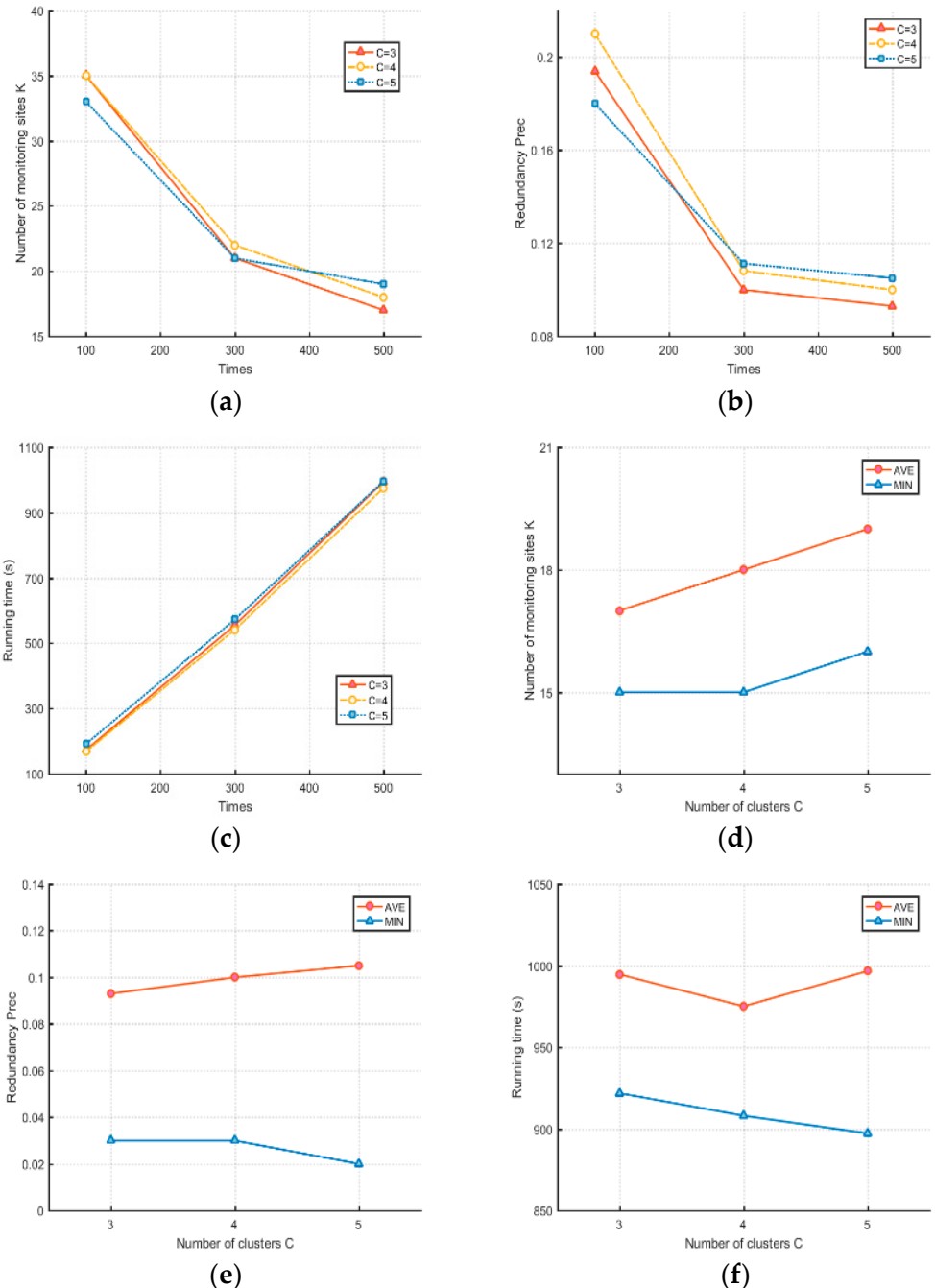

**Figure 5.** The related parameters of monitoring point deployment in different clustering bunches: (**a**) effect of the number of clusters on the number of monitoring points. (**b**) Effect of the number of clusters on redundancy. (**c**) Effect of the number of clusters on running time. (**d**) Mean vs. minimum value of the number of monitoring points for different numbers of clusters. (**e**) Mean vs. minimum of redundancy for different number of clusters. (**f**) Mean vs. minimum of running time for different numbers of clusters.

From the above Figure 5, we can receive that: (1) as the number of iterations increases, the number of monitoring points K, and redundancy Prec gradually decrease and stabilize, and the running time gradually increases, and further, the more number of clustering bunches, the slower the variation trend of monitoring points; (2) the number of monitoring points, redundancy, and running time of the three types of clusters fluctuate less after the iteration stabilized; (3) according to the mean change of the experimental results, when the number of clusters increases from 3 to 5, the number of monitoring points increase from

17 to 19, which thus increase by 7.43%; the redundancy rises from 0.093 to 0.105, which increases it by 12.5%; and the running time fluctuates by 21.6 s, which is an increase of 2.21% compared with the minimum time. Comprehensive analysis reveals that the number of clusters has a small effect on the monitoring point deployment scheme.

### 5.3. Effect of Average Importance of Monitoring Areas

The influence of importance on the program running time, number of nodes, and redundancy in this area. In disparate monitoring areas, since the importance in each grid cell is different, the distribution of weight in entire monitoring areas is also diverse, and in this paper, average importance as the parameter is utilized to monitor the overall importance distribution of the region and if the average importance is higher, it indicates that there are more important grid cells in the monitoring area and, conversely, if the average importance is lower, it suggests that there are more unimportant grid cells distributed in the monitoring area. The experiment set C = 3; $R_g$ = 3 km; $\omega_{ave}$ = 0.3/0.5/0.7. The optimal results of the experiment are shown in Figure 6a–c:

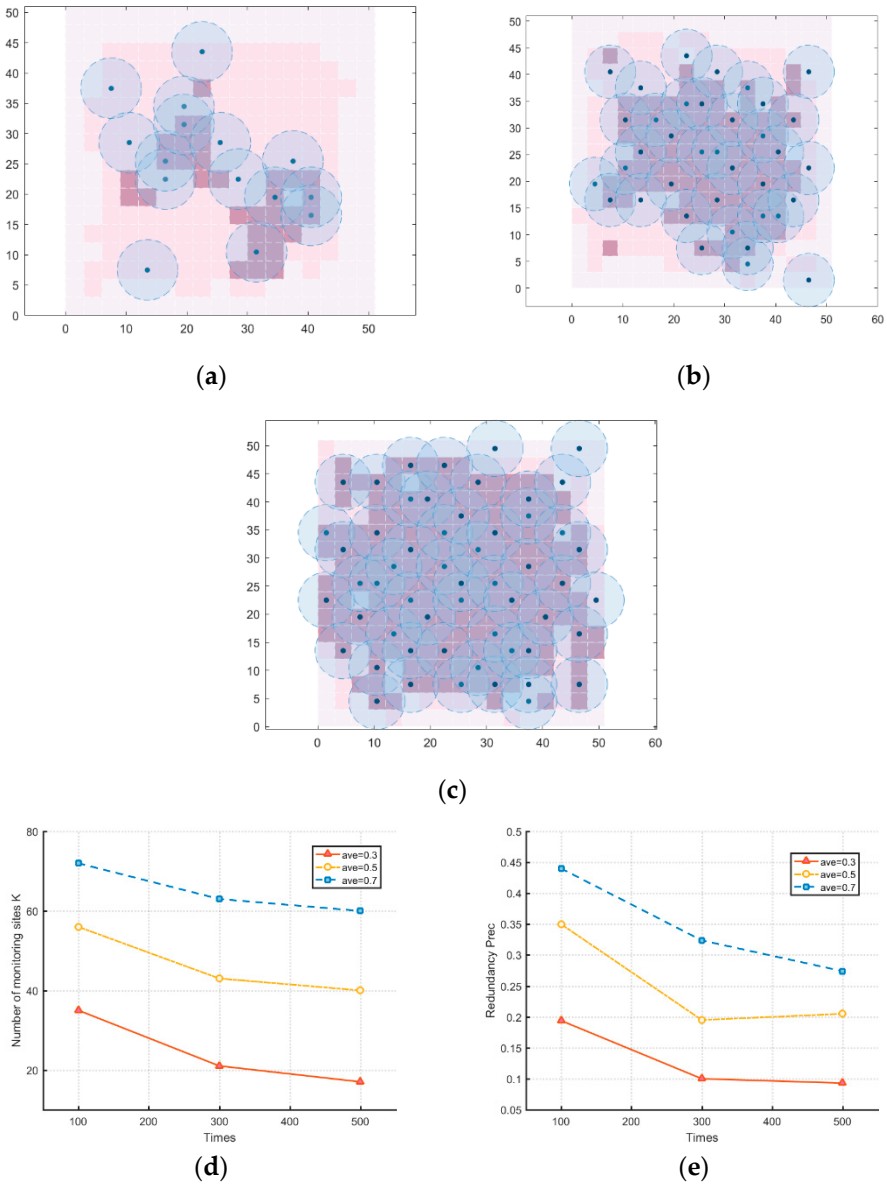

**Figure 6.** *Cont.*

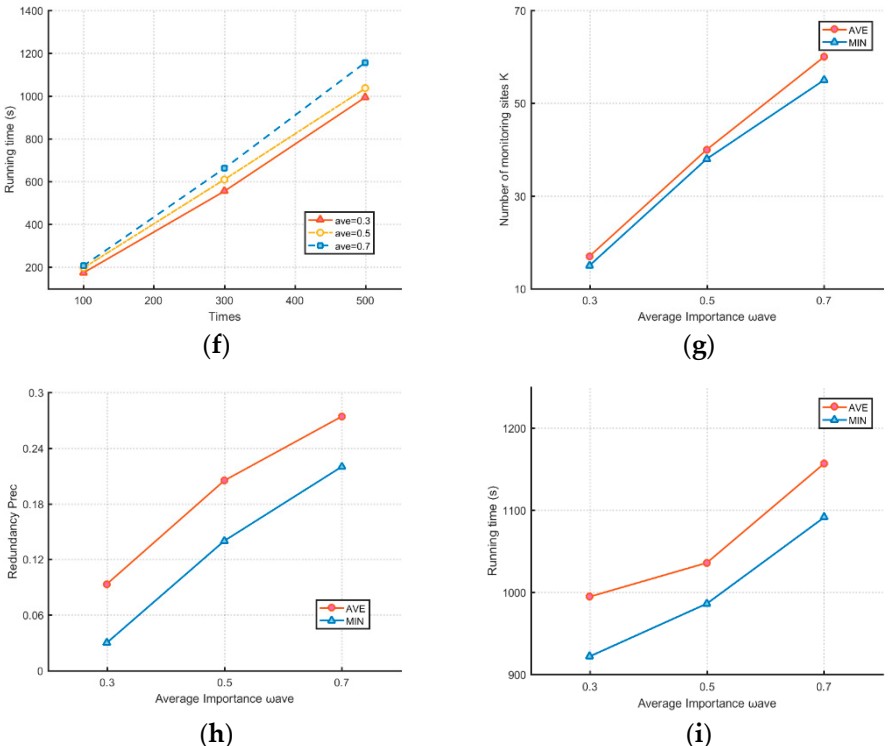

**Figure 6.** Calculated results of monitoring point deployment with different average importance and related parameters (units of Figure (**a**–**c**): km) (in Figure (**a**–**c**)): the pink grid indicates areas of different importance, the blue dots represent the location of monitoring points, and the blue circles represent the monitoring range of monitoring points): (**a**) The average importance $\omega_{ave}$ = 0.3, K = 15, Prec = 0.02, Time = 996.881 s. (**b**) The average importance $\omega_{ave}$ = 0.5, K = 38, Prec = 0.2, Time = 1036.814 s. (**c**) The average importance $\omega_{ave}$ = 0.7, K = 55, Prec = 0.22, Time = 1117.41 s. (**d**) Effect of average importance on the number of monitoring points. (**e**) Effect of average importance on redundancy. (**f**) Effect of average importance on running time. (**g**) Mean vs. minimum value of the number of monitoring points with different average importance. (**h**) Mean vs. minimum of redundancy with different average importance. (**i**) Mean vs. minimum running time for different average importance.

From the analysis of Figure 6d–i, we can obtain that: (1) as the number of iterations increases and the higher of average importance, the faster the gradient of decreasing number of monitoring points, the faster the trend of increasing running time; (2) as the iteration tends to be stabilized, the variation trends of average value and minimum value of those three indicators are basically the same, and as the increasing of average importance and the number of monitoring points, the redundancy increasing trend becomes slower, the increasing trend of running time become faster; and (3) according to the mean change of the experimental results, when the average importance increases from 0.3 to 0.7, the number of monitoring points increases from 17 to 60, with an increase of 244.57%; the redundancy increases from 0.093 to 0.274, with an increase of 193.75%; and the running time increases from 994.67 s to 1157 s, with an increase of 16.32%. It can be concluded that the average importance of the monitoring area has an essential influence on the monitoring point deployment scheme.

### 5.4. Effect of Weighted Coverage and Unweighted Coverage

Comparing the weighted area coverage algorithm with the unweighted area coverage algorithm. The algorithm proposed by this paper is compared with the deployment method of indifferent universe coverage, and in the deployment method of indifferent universe coverage, according to the coverage range of the sensor, the diamond-shaped coverage is selected and shown in Figure 7b, in order to achieve a coverage of 1 in all areas with the

minimum number of monitoring points. The experiment compared the final number of nodes, redundancy, and running time of two algorithms, and set parameters: $R_g$ = 3 km and the deployment of monitoring points based on weight is shown in a, and C = 3, $\omega_{ave}$ = 0.3; the design sketch of indifferent monitoring universe coverage is shown in Figure 7b:

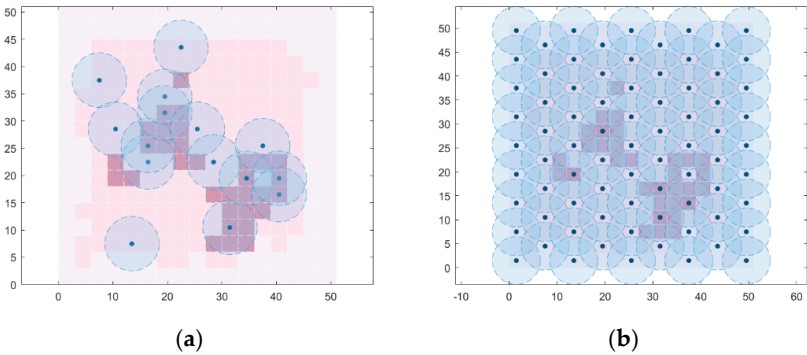

(a)          (b)

**Figure 7.** The calculation results of unweighted monitoring point deployment (units:km) (In the figure: pink grids indicate different importance areas, blue dots represent monitoring point locations, and blue circles represent monitoring range of monitoring points.): (**a**) based on weight coverage K = 15, Prec = 0.2, Time = 996.881 s. (**b**) Undifferentiated weight coverage K = 77, Prec = 0.24, Time = 6.093 s.

The comparison diagram is obtained from data and as shown in Figure 8: the weight-based coverage algorithm reduces the number of monitoring points by 80.5% on average; contrast that with indifferent coverage, while decreasing the redundancy by 87.5%, significantly increasing the utilization of monitoring points, improving monitoring efficiency, and avoiding the waste of monitoring resources. However, the weight-based coverage requires a lot of calculation time, whereas the indifferent coverage can be deployed directly.

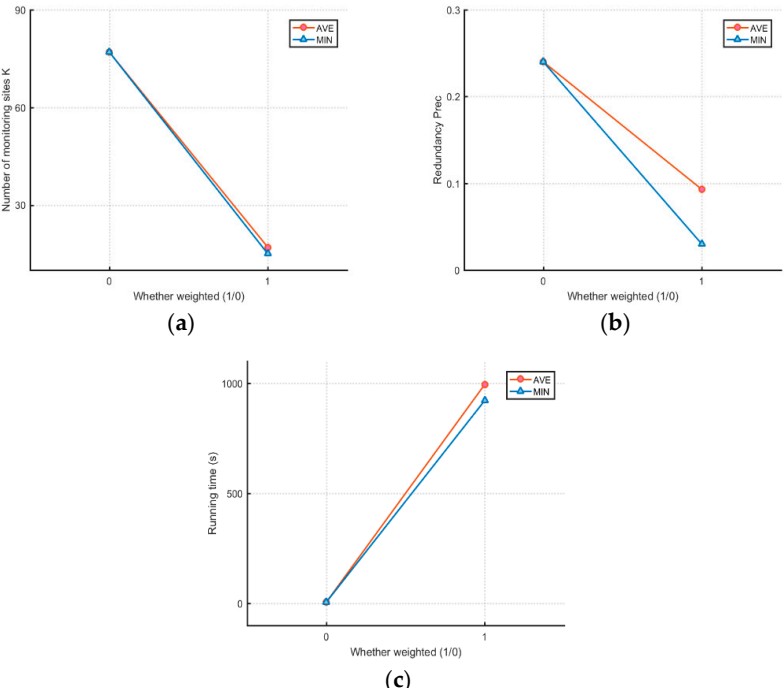

**Figure 8.** The related parameters of weighted monitoring point deployment:(**a**) Mean vs. minimum of the number of monitoring points with and without weights. (**b**) Mean vs. minimum of redundancy with and without weights. (**c**) Mean vs. minimum runtime with and without weights.

## 6. Conclusions

This paper uses the monitoring point deployment in chemical hazard areas under mesoscale as the research background, establishes a mathematical model of telemetry sensor deployment under grid division, and on this basis, constructs a weight-based model for the monitoring point deployment in chemical hazard areas, using the k-means clustering algorithm combined with the importance of grid cells, and solves the monitoring point deployment model using rapid convergent fireworks algorithm. This paper discusses the effects of different indicators on parameters such as the number of monitoring points, redundancy, and running time. Moreover, the experiment results indicate that the grid edge length divided by importance and the importance distribution situation of monitoring areas have significant influence on monitoring point deployment scheme, but the choosing of clustering number has relatively small influence. By comparing with the scheme of indifferent universe coverage, the effectiveness of the deployment model and solution algorithm is verified, and the focused problem of monitoring point deployment is solved, and the reasonable control index parameters can improve the monitoring efficiency to a certain agreement and avoid the excessive waste of deployment resources.

In our future work, we will continue our research in the following three areas.

- We can choose multiple heuristic algorithms to solve the model. In our research, we only rely on the fireworks algorithm to solve the model, and find a better algorithm to choose instead through the comparison and verification of multiple heuristic algorithms;
- We can improve the location optimization algorithm by combining it with other heuristics, such as simulated annealing, ant colony algorithm, and genetic algorithm;
- We can extend the research content to the optimization of the deployment of spatial monitoring points [55]. The location of sensors used to monitor the diffusion of chemically hazardous gases has only been studied in a two-dimensional planar area, and three-dimensional space with height has not been considered.

**Author Contributions:** Conceptualization, Y.S. (Yimeng Shi), H.Z. and Z.C.; methodology, Y.S. (Yimeng Shi) and H.Z.; software, Y.S. (Yimeng Shi); validation, Y.S. (Yimeng Shi), Z.C. and Y.S. (Yueyue Sun); formal analysis, Y.S. (Yimeng Shi) and H.Z.; investigation, Y.S. (Yimeng Shi), H.Z., Z.C., Y.S. (Yueyue Sun), X.L. and J.G.; resources, H.Z., Z.C. and J.G.; data curation, Y.S. (Yimeng Shi), Y.S. (Yueyue Sun), X.L., J.G.; writing—original draft preparation, Y.S.(Yimeng Shi) and H.Z.; writing—review and editing, Y.S. (Yimeng Shi), H.Z., Z.C. and Y.S. (Yueyue Sun); visualization, Y.S. (Yueyue Sun), X.L. and J.G.; supervision, H.Z. and X.L.; project administration, H.Z. All authors have read and agreed to the published version of the manuscript.

**Funding:** The research was financially supported by a grant from the subject (approval number 20191A060258).

**Institutional Review Board Statement:** Not applicable.

**Data Availability Statement:** Not applicable.

**Conflicts of Interest:** The authors declare no conflict of interest. The funders had no role in the design of the study; in the collection, analyses, or interpretation of data; in the writing of the manuscript; or in the decision to publish the results.

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
