# Peer review of "A Study on the Deployment of Mesoscale Chemical Hazard Area Monitoring Points by Combining Weighting and Fireworks Algorithms"

_sustainability, doi:10.3390/su15075779_

Round 1
Reviewer 1 Report
This manuscript entitled " A study on the deployment of mesoscale chemical hazard area monitoring points by combining weighting and fireworks algorithms" is a carefully done study. This work innovatively proposes a method to deploy mesoscale chemical hazard zone monitoring points by combining regional importance and fireworks algorithm. With regional importance as the constraint, a site selection model for the deployment of monitoring points in mesoscale chemical hazard zones is constructed and solved by clustering algorithm and fireworks algorithm, while relevant factors affecting the deployment scheme are discussed, and the conclusions are quite practical and instructive, which are suitable for publication in the journal "Sustainability". However, I would like to propose some details that need to be modified:
1) I suggest the author add a workflow diagram to make the article more clear;
2) I suggest the authors change the letter "K" in "K-means clustering algorithm" to lowercase;
3) A space is needed between "s,t" and the inequality in Equation 14;
4) Equation 20 numbering format is not consistent with other equation numbering format;
5) The quality of the pictures in Figure 3 and Figure 5 in the article is not clear enough, I suggest the author to improve the clarity of the pictures;
6) In line 339, there should be one space between round and (), and the font of round is suggested to be changed to the font used in the formula;
7) No spaces are needed after the keyword "telemetry", I suggest deleting it;
8) There are extra spaces in line 39 of the article, suggest deleting;
9) Suggest the author to delete the extra space between "choose" and "the" in line 126 of the article;
10) There is an extra space between "optimize" and "the" in line 139 of the article;
11) The expression format of the formula is inconsistent in the article, and it is recommended to use ":" or "." after " Equation (1)";
12) I suggest that the authors indicate the units of coordinates in Figures 1, 3, 5and 6 and specify the units of side length and running time
Author Response
Dear Reviewers,
We appreciate your professional review work on our articles. As you are concerned, there are several issues that need to be addressed. Based on your suggestions, we have made extensive revisions to the previous draft, which are described below.
Comment 1: You suggest the author add a workflow diagram to make the article more clear;
Response 1:Thanks for your great suggestion on improving the accessibility of our manuscript. We have added a workflow diagram in Figure 1 of the article.
Comment 2: You suggest the authors change the letter "K" in "K-means clustering algorithm" to lowercase;
Response 2: Thank you for your detailed comments. We have carefully and thoroughly proofread the manuscript and have corrected such issues in the full text.
Comment 3: A space is needed between "s,t" and the inequality in Equation 14;
Response 3: Thank you for your detailed comments. We have added spaces to address the issue here.
Comment 4: Equation 20 numbering format is not consistent with other equation numbering format;
Response 4:Thank you for your detailed comments. We have modified the format for the problem here.
Comment 5: The quality of the pictures in Figure 3 and Figure 5 in the article is not clear enough, I suggest the author to improve the clarity of the pictures;
Response 5:Thank you for your valuable comments. We have readjusted the clarity of the pictures for the problem of unclear pictures.
Comment 6: In line 339, there should be one space between round and (), and the font of round is suggested to be changed to the font used in the formula;
Response 6:Thank you for your comment. A correction has been made to address this issue in line 342 of the article.
Comment 7: No spaces are needed after the keyword "telemetry", I suggest deleting it;
Response 7:Thank you for your comment. A correction has been made to address this issue in line 22 of the article.
Comment 8: There are extra spaces in line 39 of the article, suggest deleting;
Response 8:Thank you for your comment. A correction has been made to address this issue in line 39 of the article.
Comment 9: Suggest the author to delete the extra space between "choose" and "the" in line 126 of the article;
Response 9:Thank you for your comment. A correction has been made to address this issue in line 129 of the article.
Comment 10: There is an extra space between "optimize" and "the" in line 139 of the article;
Response 10:Thank you for your comment. A correction has been made to address this issue in line 132 of the article.
Comment 11: The expression format of the formula is inconsistent in the article, and it is recommended to use ":" or "." after " Equation (1)";
Response 11:Thank you for your comment. A correction has been made to address this issue in line 192 of the article.
Comment 12: I suggest that the authors indicate the units of coordinates in Figures 1, 3, 5and 6 and specify the units of side length and running time
Response 12:Thank you for your comments. To address this issue we have added units and descriptions to the figure notes for figures 1, 3, 5 and 6.
We would like to take this opportunity to thank you for all your time involved and this great opportunity for us to improve the manuscript. We hope you will find this revised version satisfactory.
Sincerely,
All Authors

Reviewer 2 Report
dear author,
It is a nice manuscript provided application of firework algorithm in sensor network optimization. However, the introduction is broad and have several incoherent sentences. Please clarify and restructure the introduction.
Title of Figure 1 3 and 5 should have more information. It should explain the dots , circle and pink square. how the clustering has been done etc.
Author Response
Dear Reviewers,
Firstly, we would like to thank you for your kind letter and for reviewers’ constructive comments concerning our article .These comments are all valuable and helpful for improving our article. All the authors have seriously discussed about all these comments. According to the reviewers’comments, we have tried best to modify our manuscript to meet with the requirements of your journal.Point-by-point responses to the reviewers are listed below this letter.
Comment 1:It is a nice manuscript provided application of firework algorithm in sensor network optimization. However, the introduction is broad and have several incoherent sentences. Please clarify and restructure the introduction.
Response 1:Thank you for your detailed comments. We have carefully and thoroughly proofread the manuscript and corrected the grammar in the introduction section.
Comment 2:Title of Figure 1 3 and 5 should have more information. It should explain the dots , circle and pink square. how the clustering has been done etc
Response 2:Thanks for your great suggestion on improving the accessibility of our manuscript.We have added more information to the captions of Figures 1, 3 and 5, explaining the meaning of these dots, circles and pink squares that appear in the figures, making them easier to understand.
Thank you again for your positive comments and valuable suggestions toimprove the quality of our manuscript.
Sincerely,
All the Authors.
